# Global DEM Product Generation by Correcting ASTER GDEM Elevation with ICESat-2 Altimeter Data

Binbin Li[1,2,3,4], Huan Xie[1,6], Shijie Liu[1], Zhen Ye[1], Zhonghua Hong[5], Qihao Weng[2,3,4], Yuan Sun[1], Qi Xu[1], Xiaohua Tong[1]

[1]College of Surveying and Geo-informatics, and Shanghai Key Laboratory for Planetary Mapping and Remote Sensing for Deep Space Exploration, Tongji University, Shanghai 200092, China
[2]JC STEM Lab of Earth Observations, Department of Land Surveying and Geo-Informatics, The Hong Kong Polytechnic University, Hung Hom, Hong Kong 999077, China
[3]Research Centre for Artificial Intelligence in Geomatics, The Hong Kong Polytechnic University, Hung Hom, Hong Kong 999077, China
[4]Research Institute for Land and Space, The Hong Kong Polytechnic University, Hung Hom, Hong Kong 999077, China
[5]College of Information Technology, Shanghai Ocean University, Shanghai 201306, China
[6]Shanghai Institute of Intelligent Science and Technology, Tongji University, Shanghai 200092, China

*Correspondence to*: Huan Xie (huanxie@tongji.edu.cn)

**Abstract.** Advancements in scientific inquiry and practical applications have put forward a higher demand for the accuracy of global digital elevation models (GDEMs), especially for GDEMs whose main data source is optical imagery. To address this challenge, integrating GDEM and satellite laser altimeter data (global coverage and high-accuracy ranging) is one of the important research directions, in addition to the technological enhancement of the main data source. In this paper, we describe the datasets and algorithms used to generate a GDEM product (IC2-GDEM) by correcting ASTER GDEM elevation data with ICESat-2 altimeter data. The algorithm scheme presents the details of the strategies used for the various challenges, such as the processing of DEM boundaries, the fusion of the different data, the geographical layout of the satellite laser altimeter data. We used a high-accuracy global elevation control point dataset and multiple high-accuracy local DEMs as the validation data for a comprehensive assessment at a global scale. The results from the validation comparison present that the elevation accuracy of IC2-GDEM is evidently superior to that of the ASTER GDEM product: 1) the root-mean-square error (RMSE) reduction ratio of the corrected GDEM elevation is between 16% and 82%, and the average reduction ratio is about 47%; 2) from the analysis of the different topographies and land covers, this error reduction is effective even in areas with high topographic relief (>15°) and high vegetation cover (>60%). ASTER GDEM has been in use for more than a decade, and many historical datasets and models are based on its elevation data. IC2-GDEM facilitates seamless integration with these historical datasets, which is essential for longitudinal studies examining long-term environmental change, land use dynamics, and climate impacts. Meanwhile, IC2-GDEM can serve as a new complementary data source to existing DEMs (such as Copernicus DEM) mainly sourced from synthetic aperture radar (SAR) observation. By cross-validating qualities, filling data gaps, conducting multi-scale analyses, it can lead to more reliable and comprehensive scientific discoveries, thereby improving the overall quality and reliability of earth science research. IC2-GDEM product is openly released via https://doi.org/10.11888/RemoteSen.tpdc.301229 (Xie et al., 2024).

## 1 Introduction

High-quality digital elevation model (DEM) data are of great significance for the field of Earth science research and scientific applications, including hydrological modeling, climate change research, natural hazard assessment, and ecosystem management. For example, high-quality DEMs can improve the accuracy of watershed delineation and flood risk assessment, promoting effective water resources management and disaster preparedness (Ao et al., 2024). In climate change research, high-quality DEMs are vital for the modeling of glacier dynamics and the sea level rise impact, thereby improving predictive accuracy and informing mitigation strategies (Fan et al., 2022; Cook et al., 2012; Tran et al., 2023; Tran et al., 2024a; Tran et al., 2024b).

Airborne LiDAR/radar and high-resolution imaging systems can produce DEMs with a centimeter to sub-meter accuracy, but this kind of DEM is costly to derive. As a result, such DEMs are not extensively accessible at a global scale (Pham et al., 2018). Laser altimeters carried on satellite platforms (Schutz et al., 2005; Xie et al., 2021; Dubayah et al., 2020; Martino et al., 2019; Tang et al., 2020) such as ICESat, ICESat-2, can provide accurate elevation information around the world, and their elevation information is freely available. The accuracy of the elevation information can attain a meter, sub-meter, or even centimeter level after data refinement (Li et al., 2023c; Zhu et al., 2022; Neuenschwander et al., 2020; Fernandez-Diaz et al., 2022; Li et al., 2021). Previous studies have refined ICESat/ICESat-2 altimeter data in polar regions to produce and release polar DEM products (Shen et al., 2022; Fan et al., 2021). However, at present, it is still difficult to use satellite laser altimeter data to generate DEMs with a medium resolution at a global scale, due to the limitation of the spacing between ground tracks (Magruder et al., 2021). At a global scale, analysis of the spatial elevation at a medium resolution still relies on DEMs derived from other satellite technologies (optical imaging and synthetic aperture radar (SAR)), but the accuracy of this kind of DEM data is lower, at multiple meters or even more than 10 m (Meadows et al., 2024; Al-Areeq et al., 2023; Yue et al., 2017; Del Rosario González-Moradas et al., 2023; Hawker et al., 2019). This data characteristic is challenged by the related scientific applications with a requirement for higher and higher elevation accuracy. Therefore, improving the accuracy of this kind of DEM data has been widely focused on and studied by many scholars. In addition to technological improvement, integrating DEM with satellite laser altimeter data is one of the important research directions. Scholars have also provided important guidance and reference to this field by using linear fitting, machine/deep learning, and other methods (Li et al., 2023a; Magruder et al., 2021; Hawker et al., 2022).

Free global DEMs (GDEMs) with a medium resolution mainly include the Advanced Spaceborne Thermal Emission and Reflection Radiometer global digital elevation model (ASTER GDEM) (Abrams et al., 2015), the Shuttle Radar Topography Mission (SRTM) DEMs (Farr et al., 2007), and the Copernicus GLO-30 DEM (Fahrland et al., 2022). The data source of these DEMs can divided into two main kinds: 1) optical imaging (for the ASTER GDEM); and 2) SAR (for the SRTM DEMs and Copernicus GLO-30 DEM). Owing to the noise and anomalies resulting from the limitations inherent in optical imaging, the elevation quality of the ASTER GDEM is typically deemed to be lower than that of the other GDEMs for which the source is radar data (Meadows et al., 2024; Del Rosario González-Moradas et al., 2023; Purinton and Bookhagen, 2021), such as the

SRTM DEMs, Copernicus GLO-30 DEM. In addition, a survey of some studies showed that the role of GDEMs for which the main source is optical imaging has not been completely replaced. For example, the ASTER GDEM elevation shows a higher quality than the SRTM elevation in some mountainous areas (Li et al., 2013; Yue et al., 2017), and also shows a better quality than the Copernicus GLO-30 DEM in some areas with steep topography (Del Rosario González-Moradas et al., 2023) or high vegetation cover (Okolie et al., 2024; Huang and Yu, 2024). Furthermore, some studies (Pham et al., 2018; Yamazaki et al., 2017; Crippen et al., 2016; Franks et al., 2020; Okolie and Smit, 2022) have also reported that diversified choices and complementary use strategies for GDEMs are still favored by many scholars and users. For instance, the data sources for the void filling of the Copernicus GLO-30 DEM (currently the best available global DEM) consist of other DEMs, including the ASTER GDEM, SRTM DEMs, and several national DEMs (Del Rosario González-Moradas et al., 2023). Thus, obtaining more accurate ASTER GDEM elevation products is a great significance for advancing the research of the GDEM diversified choices and complementary use strategies. Researchers can use the enhanced ASTER GDEM in conjunction with other DEM products to cross-validate qualities, fill data gaps, and conduct multi-scale analyses (Del Rosario González-Moradas et al., 2023). This complementary use of multiple DEMs can lead to more reliable and comprehensive scientific discoveries, thereby improving the overall quality and reliability of geoscience research (Del Rosario González-Moradas et al., 2023; Ao et al., 2024). However, few previous studies have focused on integrating ASTER GDEM and satellite laser altimeter data to generate and release a new enhanced ASTER GDEM product at a global scale.

In this study, given this issue, we collected multiple sets of data with a global coverage, including ICESat-2 altimeter data, a global land-cover product, a global vegetation index product, and ASTER GDEM Version 3 data (the most up-to-date version) to correct the ASTER GDEM elevation and build a corresponding corrected product for further exploring and enhancing the applicability of the GDEMs whose main data source is optical imagery. Taking into account the particularity of polar areas, namely the high variability of ice sheets and ice flow rates, the corrected product covers the global land areas but not polar areas. Moreover, in our previous study, we presented a high-accuracy ICESat-2 elevation correction method (Li et al., 2021) and a DEM elevation correction model (Li et al., 2023a). In this study, based on these models, we further optimized and designed an automatic processing scheme for correcting ASTER GDEM elevation. The scheme does not require human intervention during the processing and is suitable for large-scale data processing. The scheme introduces detailed strategies for the various challenges, such as the processing of DEM boundaries, the fusion of different data, and the geographical layout of the ICESat-2 altimeter data. The details of the scheme will provide a meaningful reference for related fields. Meanwhile, a high-accuracy global control point dataset (elevation RMSE: 0.5-3 m within different topographies) (Li et al., 2022; Xie et al., 2021) and multiple local DEMs (LDEMs) with a high resolution and accuracy were used to validate the accuracy of the original and refined ASTER GDEM, including comparison of the elevation accuracy in areas with different geolocations, topographic relief, and vegetation cover. The related analysis results will provide a beneficial supplement for the accuracy qualification of the latest ASTER GDEM with different geolocations, altitudes, topography, and vegetation cover. Moreover, ASTER GDEM has been used for more than a decade, and many historical datasets and models are based on its elevation data. The release of

refined ASTER GDEM facilitates seamless integration with historical datasets, which is essential for longitudinal studies examining long-term environmental changes, land use dynamics, and climate impacts.

Throughout the remainder of this paper, we introduce the materials (Section II), present the methodology (Section III), and compare the accuracy of the ASTER GDEM before and after elevation correction (Section IV). Finally, we draw our conclusions in Section V.

## 2 Materials

### 2.1 Satellite Laser Altimeter Data

The ICESat-2 satellite embarked on its mission in 2018 (Martino et al., 2019). This satellite operates in a near-polar orbit, skimming the Earth at a low altitude of 496 km, featuring an orbital inclination of 92 degrees, and completing its cycle every 91 days. (Martino et al., 2019; Markus et al., 2017). The satellite (Markus et al., 2017) carries a new type of laser altimeter, i.e., a photon-counting laser altimeter, which observes global areas with three pairs of laser beams. These characteristics result in the observations at higher-latitude areas being denser than those at lower-latitude areas. Moreover, compared with previous altimeters, this altimeter only requires low laser transmission energy to observe a profile with a smaller laser footprint along the ground track.

The ATL08 product (Neuenschwander and Pitts, 2019) was used in this study, spanning a survey period from 2018 to 2022.

### 2.2. GDEM with a Medium Resolution

The ASTER GDEM Version 3 (Abrams et al., 2020) was produced through the utilization of stereo-pair images captured by the ASTER instrument onboard the Terra satellite. The DEM covers land areas between 83°N and 83°S, offering a spatial resolution of approximately 30 m. The whole DEM is split into over 20,000 files. The geo-boundary of each file is 1° × 1°. Compared with other areas, ICESat-2 laser altimeter observations are denser in polar areas, making this data particularly suitable for generating polar DEMs. Previous studies have refined ICESat/ICESat-2 altimeter data to produce and release polar DEM products. Moreover, ASTER GDEM does not fully cover polar areas, and correcting elevations in these regions is challenging due to the high variability in ice sheets and flow rates. Therefore, only the parts of the product within the land areas between 83°N and 60°S (except for polar areas) were corrected.

### 2.3. Auxiliary Data

A global land-cover product and a global vegetation index product were used to generate the evaluation attribute set for the GDEM elevation. In this study, the global land-cover product was FROM-GLC10, for which the resolution is about 10 m (Gong et al., 2019), and the global vegetation index product was GFCC30TC, which has a resolution of about 30 m (Sexton et al., 2013; García-Álvarez and Lara Hinojosa, 2022). These two data sources were matched with the ASTER GDEM by resampling.

## 2.4 Validation Data

Two different kinds of validation data were used in this study, according to the type of survey platform (i.e., satellite and airborne platforms). We adopted this strategy for validating ASTER GDEM before and after elevation correction more comprehensively, in order to reduce the impact of biased validation. The first was a high-accuracy global elevation control
point dataset (HAGECPD) (Li et al., 2022). The source of the HAGECPD data is ICESat altimeter data with a survey time between 2003 and 2009. In HAGECPD, the elevation control points cover the global land area between 83°N and 60°S. The resolution of these elevation control points is more than a hectometer along the ground track, and the distance separating the ground tracks typically measures around ~7 kilometers. Figure 1 illustrates the elevation control points distributed across a grid of 1° × 1°. The RMSE of these elevation control points is about 0.5 to 3 m. The second source of validation data was
LDEM data from around the world, for which the spatial distribution of the data is depicted in Figure 2. These LDEM data were generated from LiDAR data collected via airborne platform or images captured by unmanned aerial vehicles in conjunction with LiDAR data. Compared with the first type of validation data, these LDEM data have a high resolution and accuracy. The detailed characteristics of the LDEM data are listed in Table 1. These LDEM data were collected according to the following criteria: 1) the data were freely available; 2) the topography and land cover of these data were diverse; 3) the
geolocations of these data were diverse around the world.

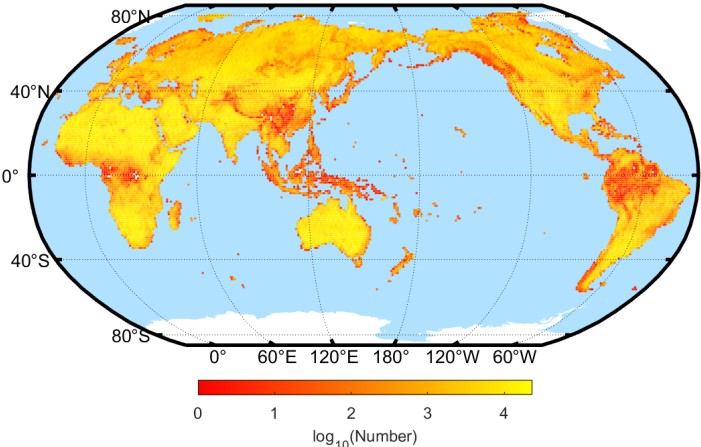

**Figure 1: Distribution of the elevation control points at 1° × 1° (redrawn from (Li et al., 2022)).**

Compared with the second type of validation data, HAGECPD has a lower data density and accuracy but a global coverage. The HAGECPD data were used to validate the accuracy of the corrected ASTER GDEM with the different geographical
locations and altitudes at a global scale. The LDEM data were used to validate the accuracy of the corrected ASTER GDEM with the different topographies and vegetation cover as they have a high data density and accuracy and their areas have diverse topography and vegetation cover.

**Table 1 Main data characteristics of the LDEMs with high resolution and accuracy**

| LDEM | Year | Resolution(m) | Vertical RMSE (cm) |
|---|---|---|---|
| a–e | 2001–2016 | 5 | ~15 |
| f | 2018–2020 | 1 | ~10 |
| g | 2018 | 1 | ~7 |
| h | 2016 | 1 | ~5 |
| i | 2020 | 0.5 | ~5 |
| j | 2017 | 0.5 | ~5 |
| k–n | 2014–2015 | 5 | ~36 to ~78 |
| o–r | 2014–2017 | | ~20 to 44 |
| s | 2020 | 1 | ~6 |
| t | 2020–2021 | 1 | ~10 |
| u | 2016–2017 | 1 | ~10 |
| v | 2016–2018 | 1 | ~10 |
| w | 2020–2022 | 0.5 | ~10 |

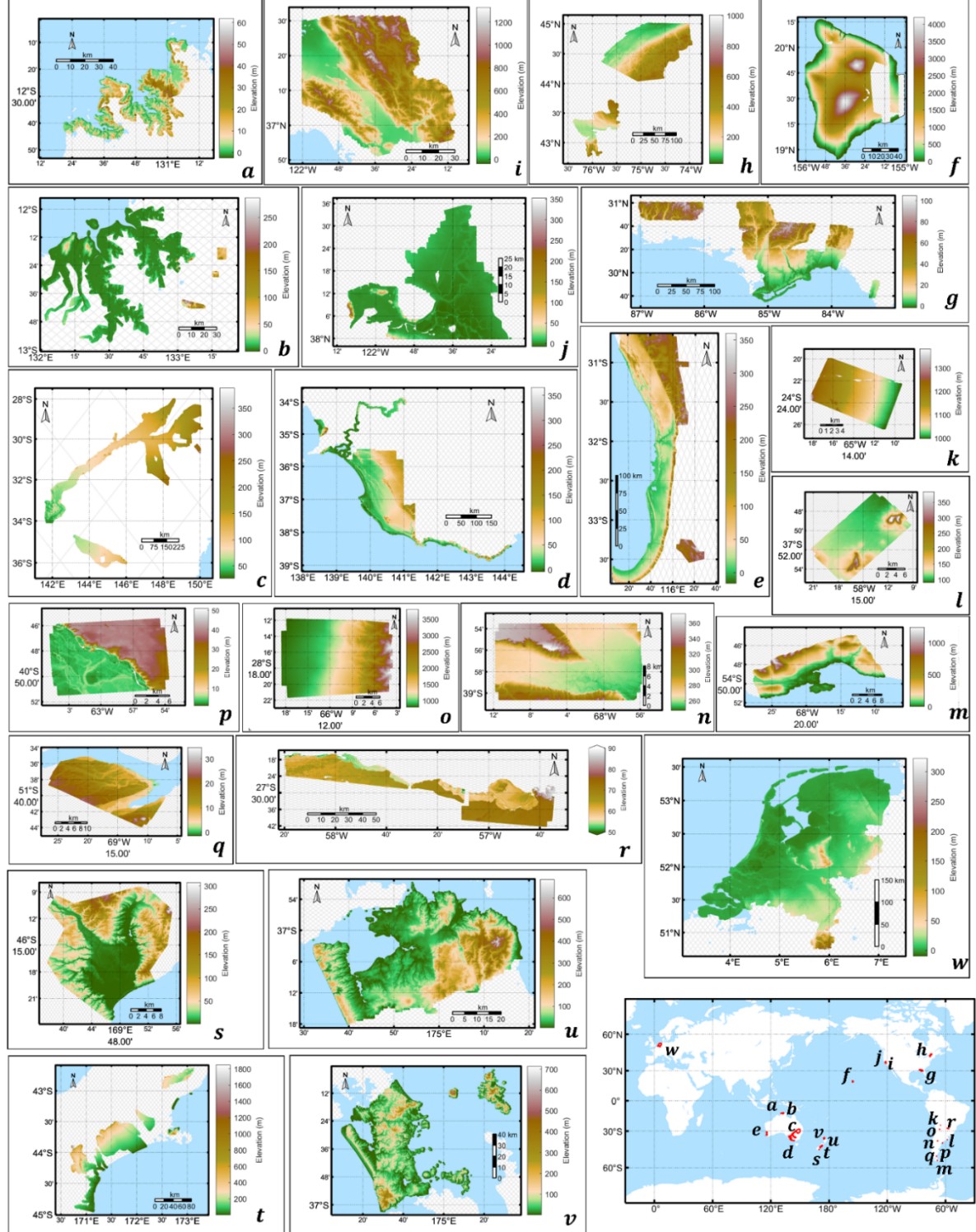

**Figure 2: Distribution of the LDEMs with high resolution and accuracy.**

## 3. Methodology

The processing flow for correcting GDEM elevation, as shown in Figure 3, includes four main parts: 1) DEM prepossessing; 2) construction of the elevation evaluation attribute set; 3) generation of the elevation deviation; and 4) DEM elevation correction.

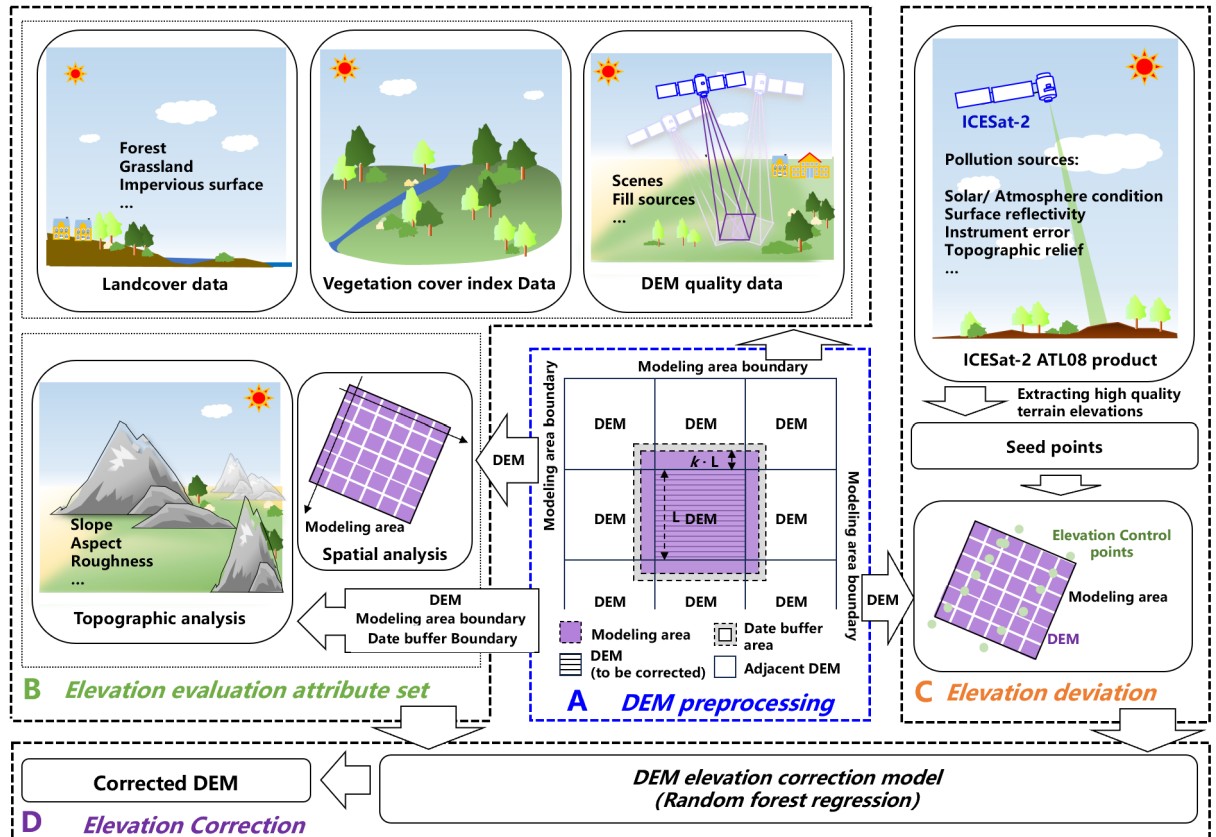

**Figure 3: Main flowchart for GDEM correction.**

### 3.1. GDEM Prepossessing

The elevations of the ASTER GDEM were corrected one by one for each 1° × 1° DEM file. In the correction, the following challenges needed to be considered: 1) there can be insufficient ICESat-2 altimeter data in some DEM files, especially in the land margin areas with a low proportion of effective land area, which can reduce the performance of the correction model within these areas; 2) elevation continuity between adjacent DEM elevations should be ensured after correction; and 3) DEM boundary pixels lack enough adjacent information, which can affect the accuracy of the evaluation attribute set of these boundary pixels. To address these challenges, it was necessary to expand the area around the DEM file to be corrected. We adopted a strategy that is to expand the boundaries of the central DEM (the processing DEM) by using data from neighboring DEMs. As shown in part A of Figure 3, the expanded area includes two types: the modeling area and the data buffer area. The

modeling area was used to address challenges 1) and 2), and the data buffer data was used to address challenge 3). The calculations for the modeling and data buffer areas were as follows:

$$\begin{bmatrix} \Phi_m^{max} \\ \Lambda_m^{min} \end{bmatrix} = R_{ref} \cdot \left( \begin{bmatrix} r_{DEM}^{ul} \\ c_{DEM}^{ul} \end{bmatrix} - k \cdot L \right) \tag{1}$$

$$\begin{bmatrix} \Phi_m^{min} \\ \Lambda_m^{max} \end{bmatrix} = R_{ref} \cdot \left( \begin{bmatrix} r_{DEM}^{dr} \\ c_{DEM}^{dr} \end{bmatrix} + k \cdot L \right) \tag{2}$$

$$\begin{bmatrix} \Phi_b^{min} \\ \Lambda_b^{max} \end{bmatrix} = R_{ref} \cdot \left( \begin{bmatrix} r_{DEM}^{dr} \\ c_{DEM}^{dr} \end{bmatrix} - k \cdot L - \Delta L \right) \tag{3}$$

$$\begin{bmatrix} \Phi_b^{min} \\ \Lambda_b^{max} \end{bmatrix} = R_{ref} \cdot \left( \begin{bmatrix} r_{DEM}^{dr} \\ c_{DEM}^{dr} \end{bmatrix} + k \cdot L + \Delta L \right) \tag{4}$$

$$L = \max \left( (r_{DEM}^{dr} - r_{DEM}^{ul}), (c_{DEM}^{dr} - c_{DEM}^{ul}) \right) \tag{5}$$

where $[r_{DEM}^{ul}, c_{DEM}^{ul}]^T$ and $[r_{DEM}^{ul}, c_{DEM}^{ul}]^T$ respectively represent the pixel coordinates of the left vertex and bottom right pixel coordinates of the DEM to be corrected; $R_{ref}$ represents the transformation matrix of pixel coordinates to geographic coordinates for the DEM to be corrected, which can be obtained directly from the DEM file; and $k$ and $\Delta L$ represent the pixel expansion coefficient and pixel expansion increment, respectively. The initial setting of $k$ was 0.1. $\Delta L$ was set to 10, which could ensure that the boundary pixels of the modeling area had sufficient adjacent information.

## 3.2. Generation of the GDEM Elevation Evaluation Attribute Set

The sources for the attribute set included the ASTER GDEM (elevation and quality data), the land-cover and the vegetation cover data. However, there were inconsistencies in the resolution of these data sources. Before constructing the attribute set, it was necessary to match each ASTER GDEM pixel with the other data based on the criterion of the nearest geographical location.

After matching, each pixel $p_i(\lambda_i, \varphi_i, h_i, q_i, \eta_i, \zeta_i)$ (called an attribute pixel) contained the geographical location (longitude $\lambda_i$ and latitude $\varphi_i$), elevation $h_i$, elevation quality $q_i$, vegetation cover index $\zeta_i$, and land-cover category $\eta_i$. Then, for each DEM of the ASTER GDEM, the corresponding attribute pixel sets $P_b$ and $P_m$ were extracted according to the boundaries of the modeling and data buffer areas, respectively, as shown in the following formulas:

$$P_b = \bigcup_{i=1}^{N} p_i \left( \lambda_i \in [\Lambda_b^{min}, \Lambda_b^{max}], \varphi_i \in [\Phi_b^{min}, \Phi_b^{max}] \right) \tag{6}$$

$$P_m = \bigcup_{i=1}^{N} p_i \left( \lambda_i \in [\Lambda_m^{min}, \Lambda_m^{max}], \varphi_i \in [\Phi_m^{min}, \Phi_m^{max}] \right) \tag{7}$$

where $N$ represents the total pixel number of the DEM to be processed and its neighboring DEM. According to the Earth's surface information, $P_b^{nw}(P_b$ within the water-free area) and $P_m^{nw}(P_m$ within the water-free area) were extracted, respectively, as shown in the following formulas:

$$P_b^{nw} = P_b \cap P_{nw} \tag{8}$$

$$P_m^{nw} = P_m \cap P_b^{nw} \tag{9}$$

$$P_{nw} = \bigcup_{i=1}^{N} pi(h_i \neq 0, \eta_i \neq 60, \zeta_i \neq 200) \tag{10}$$

where $P_{nw}$ represents the set of attribute pixels within the water-free area of the DEM to be processed and its neighboring DEM. Then, according to our previous study (Li et al., 2023a), the attribute pixels in $P_b^{nw}$ were used to generate the evaluation features for the topography, and those in $P_m^{nw}$ were used to generate the other evaluation features. The evaluation attribute set in the model includes topography, surface coverage, spatial distribution, and data-source quality (Li et al., 2023a).

### 3.3. Generation of the GDEM Elevation Deviation

The accuracy of ICESat-2 altimeter data significantly exceeds that of ASTER GDEM data. ICESat-2 altimeter data can be used as the source of the accurate elevation references (i.e., seed points) for the ASTER GDEM to generate the training samples. For the training samples, the seed points can be expected to be of better quality, followed by a more even distribution, and of greater quantity. However, there is a contradiction between these indicators. On a global scale, it was necessary to make comprehensive consideration of these indicators in each DEM area of the ASTER GDEM. For this comprehensive consideration, in our prior study, we presented a method for deriving accurate elevations from ICESat-2 altimeter data with the comprehensive evaluation labels. We then analyzed the performance of each evaluation label, and discussed a balanced strategy between the quality and distribution/quantity of the extracted elevations (Li et al., 2021). Here, based on our previous study, we extracted two types of seed points from the ICESat-2 altimeter data through different combinations of evaluation labels. The first type of seed point (FYSP) was extracted by a comprehensive evaluation label to ensure as high an elevation quality as possible. The second type of seed point (SYSP) were extracted by partial assessment labels (outliers and atmospheric assessments) to ensure good elevation quality and good quantity/distribution. According to the two types of seed points, there are two strategies for the adjustment of the adoption: 1) the adjustment strategy of seed point number (ASSPN); and 2) the adjustment strategy of seed point distribution (ASSPD).

For ASSPN, it was necessary to consider two situations for each $1° \times 1°$ DEM: 1) the quantity of ICESat-2 altimeter data increases with the increase of latitude; and 2) the effective land area is different at the same latitude. To this end, the type of seed point to adopt was related to the maximum latitude $\Phi_m^{max}$ and effective land proportion of the modeling area $r_{land}$, as shown in the following equations. If the number of FYSP is higher than $N_S$, FYSP was adopted; otherwise, SYSP was adopted.

$$N_S = r_{land} \cdot cos(\Phi_m^{max}) \cdot N_o \tag{11}$$

$$r_{land} = \frac{|P_m^{nw}|}{|P_m|} \tag{12}$$

ASSPD can be expected to obtain a better spatial distribution of seed points and is a supplement for ASSPN. ASSPD is used for a situation in which ASSPN adopts FYSP as SYSP has a good quantity/distribution. ASSPD quantifies the distribution of

the first type of seed point by dividing each modeling area into sub-regions and then uses the second type of seed point to
replace the regions with a low number of seed points. As shown in the following formulas, each modeling area was evenly
divided into sub-regions ($30 \times 30$ was adopted in this study) to generate a set $P_I$ that a statistic results of the number of the
seed points in each sub-region.

$$P_I = \bigcup_{i=1}^{n_I} (\Gamma_i) \tag{13}$$

$$\Gamma_i = \sum_{j=1}^{n} \left\| \frac{\gamma_j^i}{\|\gamma_j^i\|} \right\| \tag{14}$$

$$\gamma_j^i = \tau_j \left( \lambda_j \in [\Lambda_i^{min}, \Lambda_i^{max}), \varphi_j \in [\Phi_i^{min}, \Phi_i^{max}) \right) \tag{15}$$

$$\Lambda_i^{min} = \Lambda_m^{min} + (i-1) \cdot \frac{\Lambda_m^{max} - \Lambda_m^{min}}{\sqrt{n_I}} \tag{16}$$

$$\Lambda_i^{max} = \Lambda_m^{min} + i \cdot \frac{\Lambda_m^{max} - \Lambda_m^{min}}{\sqrt{n_I}} \tag{17}$$

$$\Phi_i^{min} = \Phi_m^{min} + (i-1) \cdot \frac{\Phi_m^{max} - \Phi_m^{min}}{\sqrt{n_I}} \tag{18}$$

$$\Phi_i^{max} = \Phi_m^{min} + i \cdot \frac{\Phi_m^{max} - \Phi_m^{min}}{\sqrt{n_I}} \tag{19}$$

After obtaining $P_I$, the median $N_{median}$(except for zeros, thus reducing the influence of the blank regions of seed points) was
calculated. The parts $P_{add}$ from which the number of seed points was less than half of $N_{median}$ were then identified, as shown
below. Finally, in the sub-regions belonging to $P_{add}$, SYSP was replaced with FYSP.

$$N_{median} = M\left( P_I(\Gamma_i \neq 0, i = 1,2 \dots N_I) \right) \tag{20}$$

$$P_{add} \in \left\{ P_I < \frac{N_{median}}{2} \right\} \tag{21}$$

where $M(\cdot)$ represents the median calculation.

Finally, all of the collected seed points matched the ASTER GDEM elevations within the modeling area, based on the criterion
of the nearest geographical location, and the matched elevations were used to obtain the elevation deviations of the ASTER
GDEM.

**3.4. GDEM Elevation Correction**

For each DEM file of the ASTER GDEM, a random forest regression algorithm was used to train its elevation correction
model. The random forest regression function (fitrensemble) of the MATLAB platform was directly used for its processing
efficiency and compatibility. The function can be viewed and downloaded via
https://ww2.mathworks.cn/help/stats/fitrensemble.html. The method of function was selected to bootstrap aggregation

(bagging, random forest) (Breiman, 2001). For this method, we adopted the recommended/default setting values for the parameter selection, including the tree number (100), learners (tree), etc. Moreover, when the number of training samples is too low, it needs to expand the area (i.e., the modeling area) for selecting training samples in order to ensure that there are sufficient training samples. If this number was less than 100, it was necessary to expand the modeling area by adjusting $k$, as shown in the following formula, and then repeat from part A in Figure 3, i.e., "DEM Prepossessing".

$$k = min(\{0.1, 0.2, \dots 1\}, L + 2 \cdot (L - \Delta L)) \tag{22}$$

After obtaining the model, all the DEM elevations (i.e., z component) were corrected by the model, and then a new DEM file was generated with the same format.

## 4. Results and Discussion

### 4.1. Release of the Corrected ASTER DEM Product

Based on the presented scheme, we corrected the ASTER GDEM elevation with ICESat-2 altimeter data and then stored the corrected ASTER GDEM product (IC2-GDEM) in the GeoTIFF format (.tif) with the same projection and datum as the ASTER GDEM (Xie et al., 2024). The resolution of IC2-GDEM is about 30 m (grid). The survey date of its main source data is between 2000-2013, and its elevation correction source is from ICESat-2 laser altimeter data with the survey date from 2018-2022. More details about the data source of the IC2-GDEM are listed in Table 2. The IC2-GDEM product has been openly released via the National Tibetan Plateau Data Center (DOI: 10.11888/RemoteSen.tpdc.301229).

Table 2 Characteristics of input and output data

| Data Type | Data name | Resolution | Description |
|---|---|---|---|
| DEM data | ASTER GDEM V3 | ~30 m (grid) | Input data, main source data, survey date: 2000 - 2013 |
| Satellite laser altimeter data | ICESat-2 ATL08 | ~100 m (along the ground track) | Input data, main source data, survey date: 2018 - 2022 |
| Landcover data | FROM-GLC10 | ~10 m (grid) | Input data, auxiliary data, survey date: 2017 |
| Vegetation cover index data | GFCC30TC | ~30 m (grid) | Input data, auxiliary data, survey date: 2015 |
| DEM data | IC2-GDEM | ~30 m (grid) | Output data |

Figure 4 shows the distribution of the IC2-GDEM product. Figure 5 shows the average of elevation corrections of IC2-GDEM relative to ASTER GDEM within the global 1°×1°grid. IC2-GDEM covers the global land area between 83°N and 60°S, except for the polar regions. Under this coverage, approximately 99.98% of the ASTER GDEM elevation has been corrected, and the

remaining ASTER GDEM elevation has not been corrected mainly because of the lack of sufficient ICESat-2 seed points, i.e. the ICESat-2 seed points did not satisfy the constraints of the presented methodology. The uncorrected ASTER GDEM elevation areas are mainly parts of islands and reefs where the geolocation is near low-latitude regions.

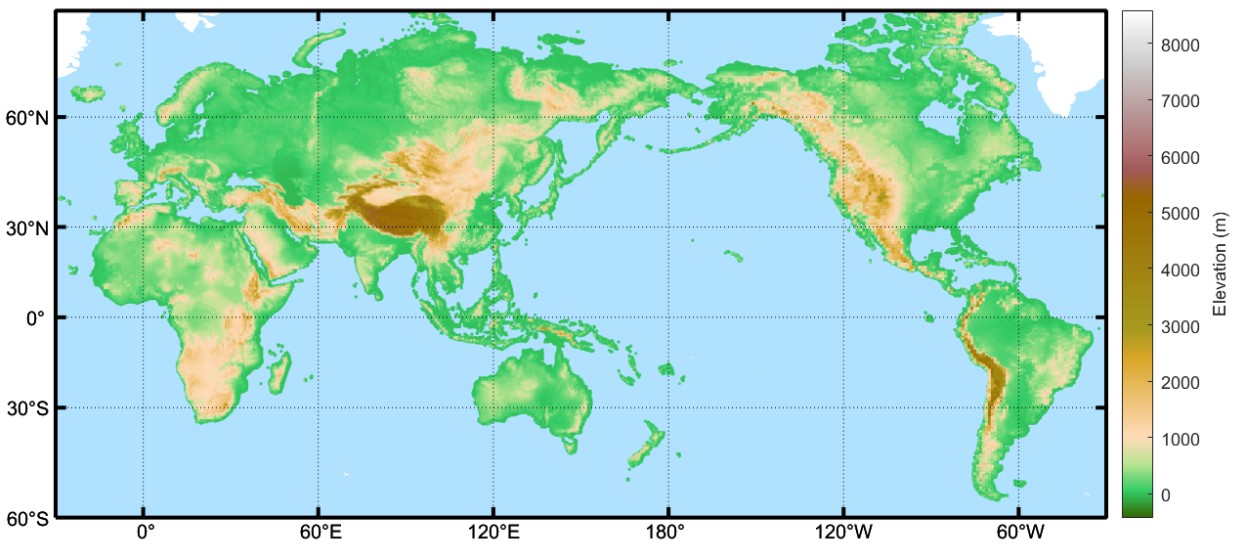

**Figure 4: ASTER GDEM product after ICESat-2 altimeter data correction.**

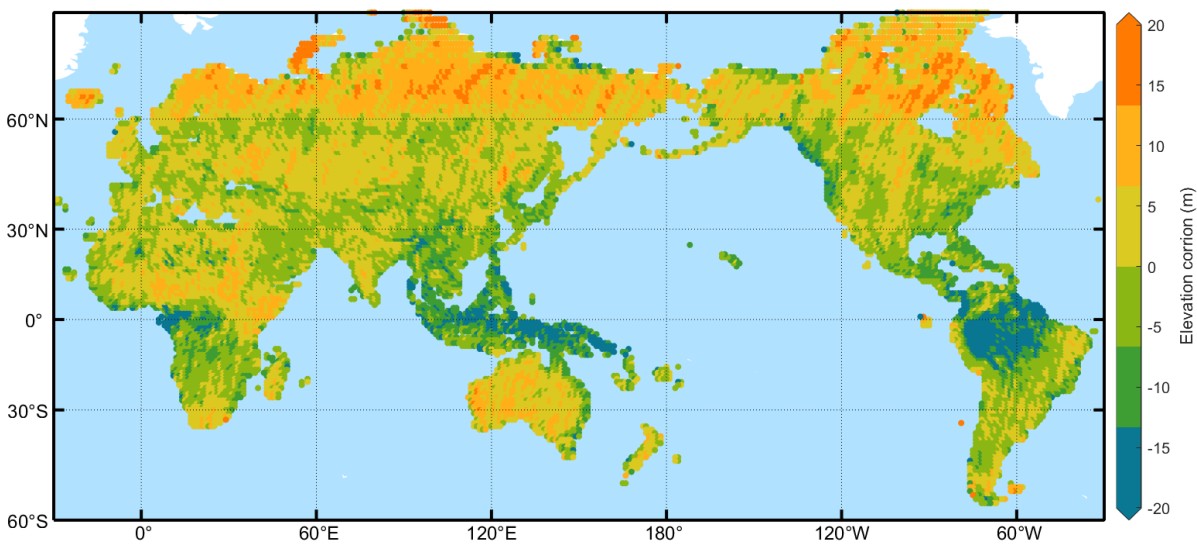

**Figure 5: Average of elevation corrections of IC2-GDEM relative to ASTER GDEM within the global 1°×1°grid.**

### 4.2. Validation by the High-quality Elevation Control Point Dataset at a Global Scale

By using the HAGECPD data (survey date: 2003-2009), we validated and then contrasted the accuracy of the original and
corrected ASTER GDEM elevation within various continents, including Asia, Africa, South America, Oceania, North America,
and Europe. Figure 6 provides the statistic results.

The results from the comparison indicate that the accuracy of the corrected ASTER GDEM elevation is higher than that of the
original GDEM elevation. In the different continents, the RMSEs of the corrected ASTER GDEM elevation are about 2 to
4.5 m, and those of the original ASTER GDEM elevation are about 8 to 10 m. After the elevation correction, the errors of the
ASTER GDEM elevation are reduced by more than 45%, and the maximum reduction ratio exceeds 70%. There are two main
reasons for this the inconsistency in elevation accuracy improvement among the different continents. The first is that there are
significant differences in the quality of ASTER GDEMs across continents. The second is that the topographic relief and the
landcovers across continents are different. These influencing factors are also important evaluation attributes in the correction
model of DEM elevation. Moreover, there is an obvious difference in the corrected ASTER GDEM elevation errors for the
different continents. The elevation error for the continents in order from lowest to highest is: Oceania, South America, Africa,
North America, Asia, and Europe. The order of the original ASTER GDEM elevation errors is similar to the above order. To
further evaluate the correction performance for the various geolocations, we scrutinized the accuracy of the original and
corrected ASTER GDEM elevations in different longitude and latitude regions. Results from the scrutiny are displayed in
Figure 7 and Figure 8.

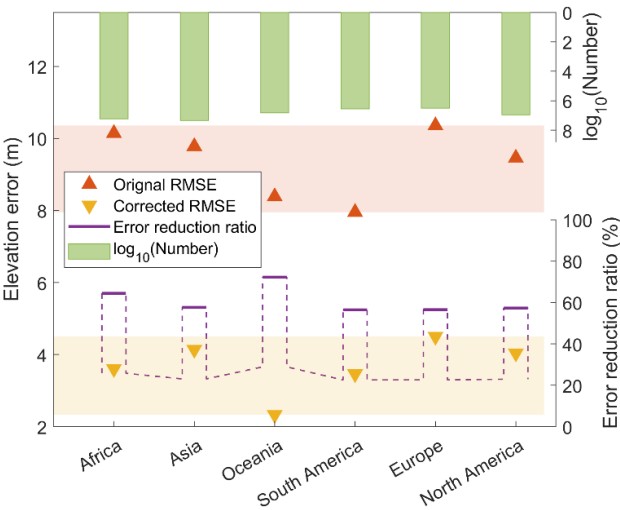

**Figure 6: Comparison of the accuracy of the original and corrected ASTER GDEM elevation within different continents.**

From Figure 7, in the different longitude regions, it is apparent that, compared with the original ASTER GDEM elevation, the
systematic error and standard deviation after ASTER GDEM elevation correction are significantly reduced. In particular, for
the original ASTER GDEM elevation, there is a large difference in the systematic error. The systematic error of the original

ASTER GDEM elevation in the low-longitude regions is smaller than that in the high-longitude regions. The systematic error after ASTER GDEM elevation correction is close to zero in the different longitude regions.

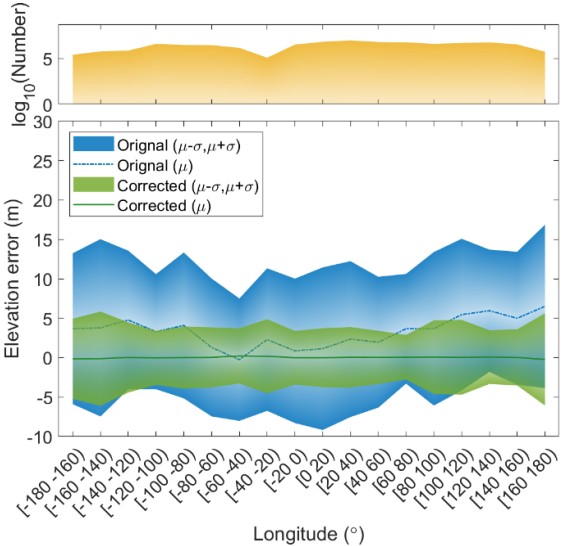

**Figure 7: Comparison of the error of the original and corrected ASTER GDEM elevation within different longitude regions.**

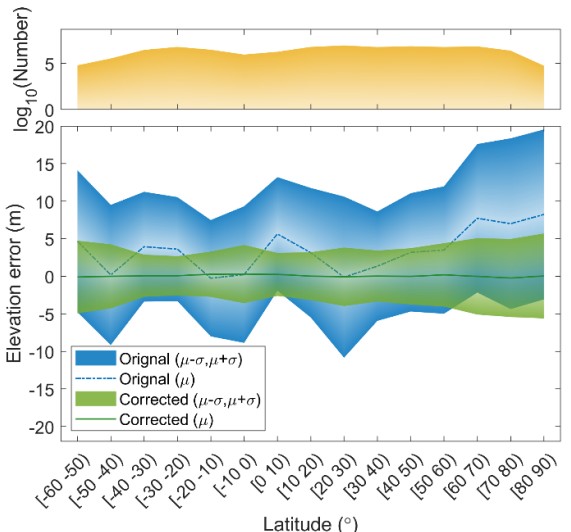

**Figure 8: Comparison of the error of the original and corrected ASTER GDEM elevation within different latitude regions.**

Similar to the analysis results in Figure 7, in the different latitude regions, it is also apparent that the systematic error and standard deviation after ASTER GDEM elevation correction are significantly reduced, compared with the original ASTER GDEM elevation, as displayed in Figure 8. Notably, the data density of the ICESat-2 laser altimeter data varies greatly in different latitude regions, and the data density of the ICESat-2 laser altimeter data in the low-latitude regions is generally less

than that in the high-latitude regions (Neuenschwander and Pitts, 2019; Markus et al., 2017). However, the errors of the ASTER

GDEM elevation corrections in low-latitude regions (especially near the equator) are no larger than those in other latitude regions, which indicates that the correction results are less affected by the difference in the data density of the ICESat-2 laser altimeter data. There are two main reasons for the observed phenomenon in DEM elevation correction. First, the DEM elevation correction model generalizes well across various spatial distributions (Li et al., 2023). Second, multiple strategies

have been designed for extracting seed points from ICESat-2 laser altimeter data during the generation of GDEM elevation deviation. These strategies ensure an even distribution of seed points, thereby mitigating the impact of varying densities in the ICESat-2 laser altimeter data. Moreover, the errors of the ASTER GDEM elevation corrections are larger in high-latitude regions, especially in the Northern Hemisphere. The primary cause for this phenomenon is that the original ASTER GDEM elevation errors in the high-latitude regions are significantly larger than those in other regions.

## 4.3. Validation by LDEM

By using the LDEM data from around the world (as shown in Figure 2), we validated the errors of the ASTER GDEM elevation before and after correction and then qualified the error reduction ratio of the corrected ASTER GDEM elevation. Table 3 displays the results.

**Table 3 Comparison of the error of the original and corrected ASTER GDEM elevation in the different LDEM areas**

| Area | Original RMSE (m) | Corrected RMSE (m) | Error reduction ratio (%) |
|------|------------------|--------------------|--------------------------|
| a | 10.69 | 3.71 | 65.33 |
| b | 7.01 | 3.04 | 56.68 |
| c | 9.33 | 1.63 | 82.49 |
| d | 10.73 | 3.53 | 67.14 |
| e | 7.27 | 3.44 | 52.74 |
| f | 11.68 | 6.33 | 45.82 |
| g | 8.46 | 4.34 | 48.73 |
| h | 7.63 | 5.16 | 32.38 |
| i | 11.09 | 7.44 | 32.95 |
| j | 5.46 | 1.83 | 66.51 |
| k | 3.82 | 3.15 | 17.55 |
| l | 7.22 | 4.57 | 36.78 |
| m | 19.37 | 12.51 | 35.42 |
| n | 7.97 | 4.26 | 46.53 |
| o | 7.71 | 6.47 | 16.14 |
| p | 11.81 | 4.36 | 63.04 |
| q | 5.68 | 3.54 | 37.71 |

| | | | |
|---|---|---|---|
| r | 5.98 | 1.88 | 68.57 |
| s | 9.56 | 5.81 | 39.26 |
| t | 11.50 | 5.15 | 55.23 |
| u | 12.56 | 7.19 | 42.79 |
| v | 13.21 | 9.99 | 24.39 |
| w | 8.56 | 2.14 | 75.04 |
| All | 9.38 | 3.37 | 64.05 |

From Table 3, it can be seen that the errors of the ASTER GDEM elevation after correction are significantly reduced. For all of the validation areas, the average RMSE of the original ASTER GDEM is 9.38 m, that of the corrected ASTER GDEM is 3.37 m, and the reduction ratio of the elevation error of the corrected ASTER GDEM is 64.05% when compared with the original ASTER GDEM. For the individual validation areas, the RMSEs of the original ASTER GDEM are between 3.82 and

335 19.37 m, and those of the corrected ASTER GDEM are between 1.63 and 12.51 m. Compared with the original ASTER GDEM, the reduction ratio of the elevation error of the corrected ASTER GDEM ranges from 16.14% to 82.49%, corresponding to an average of 47.66%.

Figure 9 provides a more detailed evaluation of the error distribution of the original and corrected ASTER GDEM elevation. In each plane, the red and orange histograms represent the error distribution of the original and corrected ASTER GDEM,

respectively. From Figure 9, it can be seen that the error distributions of the corrected ASTER GDEM elevation show better symmetry at zero error than those of the original ASTER GDEM elevation. This phenomenon indicates that the systematic error of the original ASTER GDEM elevation has been well corrected. Meanwhile, the statistical dispersion of the error distribution of the corrected ASTER GDEM elevation is significantly smaller than that of the original ASTER GDEM elevation. Moreover, there are differences between these statistical descriptions. This phenomenon is mainly caused by differences in

topography and the Earth's surface. To confirm this, we combined other data, including LDEM data, vegetation cover index data (GFCC30TC), and land-cover data (FROM-GLC10), to evaluate the influence of these two factors. Results from the analysis are provided in Figure 10 and Figure 11. For terrain, we used the commonly used topographic indicator—slope—to quantify the topographic relief. The slope was obtained through ArcGIS processing. After the slope was obtained, we first divided it into different slope intervals and then calculated the errors after ASTER GDEM elevation correction in each interval.

Meanwhile, the errors of the original ASTER GDEM elevation were added to the statistical analysis to further supplement the investigation of the correction results. To examine the different surfaces of Earth, we first divided it into urban, bare land, and vegetation areas, according to the land-cover data, and then carried out statistical analysis of the errors of the original and corrected ASTER GDEM elevation in each area. We then carried out subdivision and statistical analysis of the vegetation areas through the vegetation cover data.

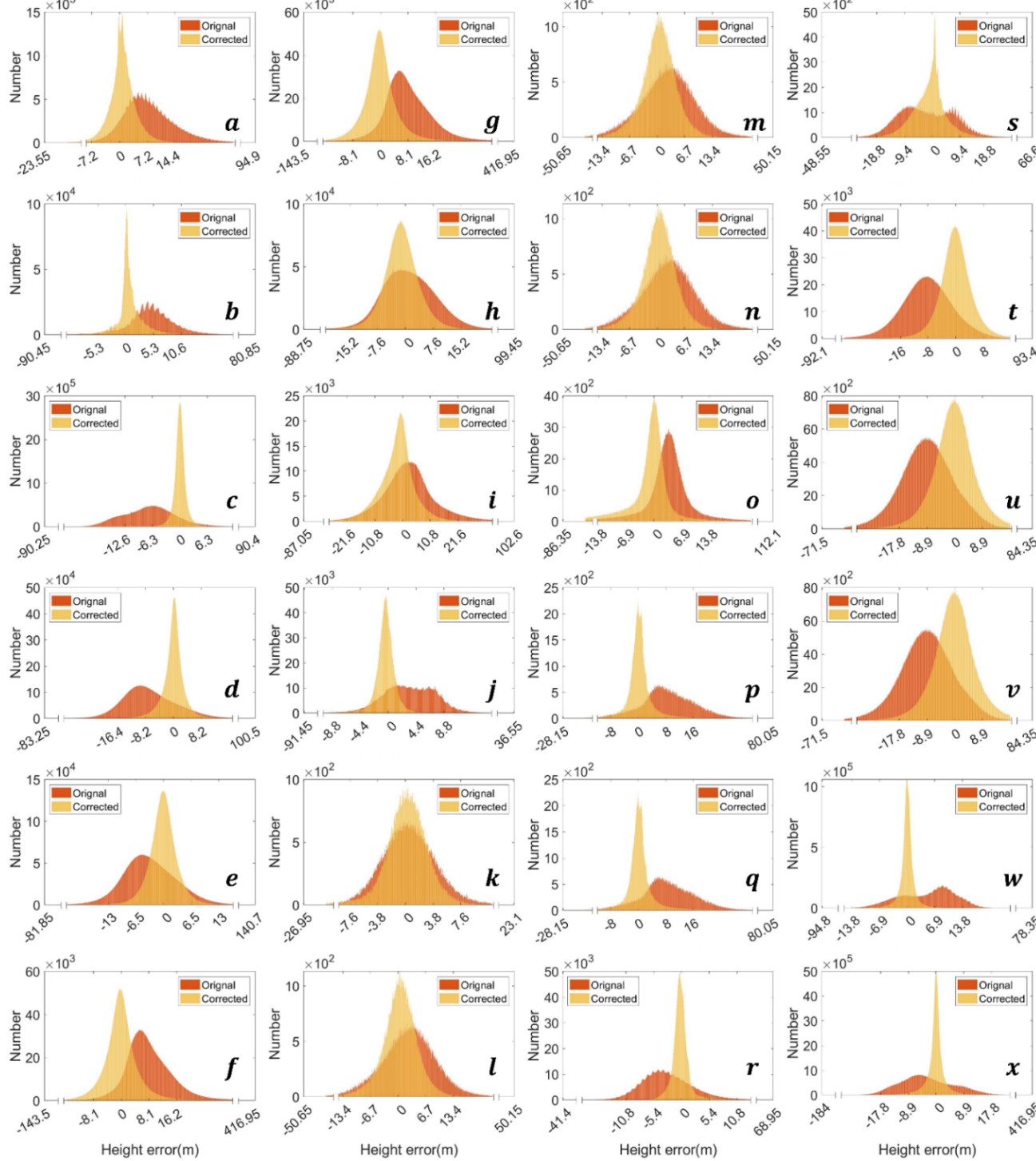

**Figure 9: Comparison of the original and corrected ASTER GDEM elevation error.**

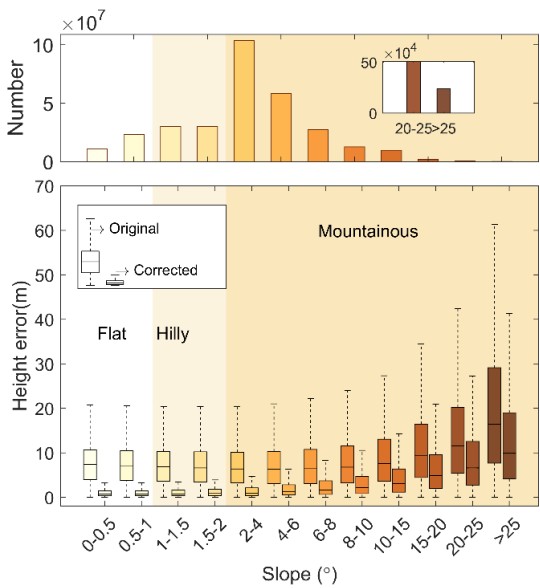

**Figure 10: Comparison of the original and corrected ASTER GDEM elevation error with different slopes.**

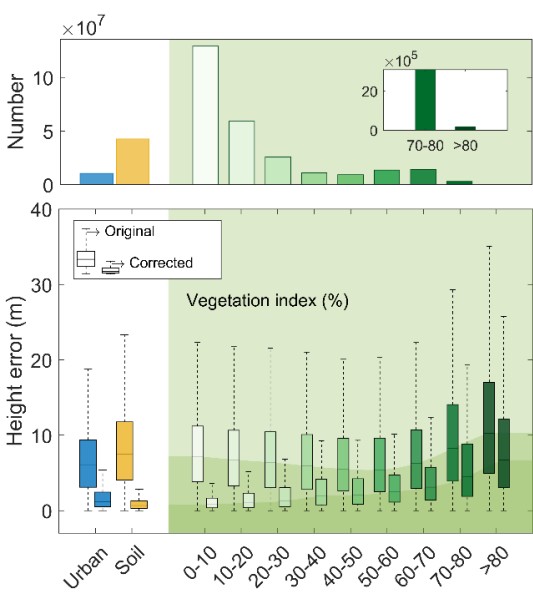

**Figure 11: Comparison of the original and corrected ASTER GDEM elevation error with different land covers.**

From Figure 10, it is apparent that, in the different slope regions, the errors of the corrected GDEM elevation are significantly reduced when compared with those of the original GDEM elevation, which indicates that the corrected results have a better elevation quality in the areas with different topographic relief. Meanwhile, this kind of error decrease shows a weakening trend with the increase of the slope. There are two main reasons for this trend. The first reason is that the quality of data from the

ICESat-2 laser altimeter, which were used as the seed points, generally decreases as the slope increases. The second reason is that the accuracy of the original ASTER GDEM elevation also decreases as the slope increases, as shown in Figure 10.

From Figure 11, under the three different Earth surface categories, it can be seen that the errors of the corrected GDEM elevation are all significantly smaller than those of the original GDEM elevation. For the different land covers, the corrected
GDEM elevation in the bare land areas shows the best elevation correction. This means that the DEM elevation correction model can reduce the impact caused by the inherent noise of optical data. Compared with the other two types of ground objects, the bare land area is not shielded by ground objects, which is conducive to optical observation of ground texture (Li et al., 2023a; Pham et al., 2018) and laser detection of ground elevation (Li et al., 2023b; Neuenschwander and Pitts, 2019). Therefore, there is little need to consider the influence of ground objects on terrain elevation in bare land areas. For the other two types
of land cover, the corrected GDEM elevation quality in the areas with low vegetation cover (less than 20% vegetation cover index) is comparable to that in the urban areas, but with the increase of vegetation cover, the difference in the corrected GDEM elevation quality between the two types of land cover gradually increases. Moreover, relative to the original GDEM elevation, the quality improvement of the corrected GDEM elevation shows a reduced trend with the increase in vegetation cover. Similar to the analysis of topographic relief, there are two main reasons for this phenomenon, i.e., the quality of the ICESat-2 laser
altimeter data and the accuracy of the original ASTER GDEM elevation both decrease as the vegetation cover increases.

The absence of LDEM data in Asia and Africa (difficult to obtain for free) is a noteworthy consideration. The lack of such data in these continents could introduce uncertainties, as the validation relies on the available LDEM data. To reduce the impact of these uncertainties, we also used the other satellite laser altimeter data with global coverage, i.e., ICESat altimeter data, as another source of the validation data in this study. The resolution and accuracy of the ICESat altimeter data may not
match those of LDEM data. This means that the validation of finer topographic details in Asia and Africa may not fully account. Despite this gap, the similar validation accuracy between the two data is advantageous, particularly given the average reduction ratios close to 47%. This similarity suggests that ICESat altimeter data can serve as a reliable alternative and a beneficial supplement for areas lacking LDEM data.

## 4.4 Limitations and Future Work

From the above analysis, while the IC2-GDEM demonstrates significant improvements in elevation accuracy, its quality of elevation correction is notably limited in the areas with steep slopes or high vegetation cover index. For research or applications with high-quality elevation requirements, this limitation can be quantified and identified through topographic relief calculations or in combination with vegetation cover data. Moreover, the temporal differences between the ICESat-2 data survey and ASTER GDEM collection may lead to elevation inconsistencies, especially in extremely dynamic landscapes (e.g.,
coastal erosion areas). To this end, for the validation areas, we selected multiple areas located near the coast for further evaluating the performance of DEM elevation correction in this kind of landscape, including the coast of the Netherlands, the coast of Australia, the islands of New Zealand, the west coast of the United States and the island of Hawaii, etc., as shown in figure 2. Validation results (Figure 9) show that the corrected DEM elevations still exhibit significant accuracy improvements.

This phenomenon is because the temporal discrepancy between the datasets is generally smaller than the ASTER GDEM elevation error (about 10 m). To further mitigate these inconsistencies, two main strategies are proposed. First, using repeated ICESat-2 observations to analyze areas with potential elevation changes can assess the application potential of IC2-GDEM in dynamic landscapes. Second, integrating IC2-GDEM with other elevation data collected closer to the ASTER GDEM collection time can reduce analysis deviations in dynamic landscapes. In future work, we will identify areas with elevation changes using the above strategies and integrate other GDEMs to further enhance IC2-GDEM elevation quality in dynamic landscapes.

## 5. Conclusion

In this study, we introduce a new open-source dataset, named IC2-GDEM. IC2-GDEM is generated by directly refining the ASTER GDEM elevation with ICESat-2 altimeter data, as well as FROM-GLC10 data and GFCC30TC data. This strategy provides a cost-effective way to improve elevation accuracy, especially beneficial for developing regions with limited resources for high-accuracy DEMs. After validating at a global scale, IC2-GDEM presents a superior elevation quality and application potentials: 1) be less affected by the difference in the data density of the ICESat-2 laser altimeter data; 2) an obvious improvement of elevation quality for the different topographies, even in the mountainous areas with higher slopes; 3) a clear improvement in elevation quality was observed across various land covers, including areas with dense vegetation cover; 4) be expected to promote seamless integration with the historical datasets of ASTER GDEM for studying longitudinal studies of long-term environmental changes, land use dynamics, and climate impacts; 5) as a new complementary data source for other GDEMs, such as SRTM, and Copernicus DEM to cross-validate qualities, fill data gaps and conduct multi-scale analyses.

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

**Data availability**

All the data used in this study are open source. The IC2-GDEM is available via https://doi.org/10.11888/RemoteSen.tpdc.301229 (Xie et al., 2024). The ICESat-2 ATL08 product is available via https://lpdaac.usgs.gov/. The ASTER GDEM product is available via https://lpdaac.usgs.gov/. The global land-cover data (FROM-GLC10) are available via http://data.ess.tsinghua.edu.cn/fromglc10_2017v01.html. The vegetation cover index data (GFCC30TC v003) are available via https://lpdaac.usgs.gov/products/gfcc30tcv003/. The HAGECPD data are available via

10.11888/Geogra.tpdc.271727. The high-resolution and accurate LDEM data are available via https://data.linz.govt.nz/, https://www.noaa.gov/, https://www.ahn.nl/, https://www.ign.gob.ar/, and https://ecat.ga.gov.au/.

**Funding**

This research was supported by the National Natural Science Foundation of China (grant no. 42325106, 42221002), the Shanghai Academic Research Leader Program (grant no. 23XD1404100), and the Fundamental Research Funds for the Central Universities of China.

**Author contributions**

Binbin Li: Writing – original draft, Visualization, Methodology. Huan Xie: Conceptualization, Writing – review & editing, Funding acquisition. Shijie Liu: Investigation, Resources. Zhen Ye: Investigation, Resources. Zhonghua Hong: Investigation. Qihao Weng: Investigation. Yuan Sun: Data curation, Validation. Qi Xu: Software, Validation. Xiaohua Tong: Funding acquisition, Supervision.

**Competing interests**

The contact author has declared that none of the authors has any competing interests.