# Peer review of "Global DEM Product Generation by Correcting ASTER GDEM Elevation with ICESat-2 Altimeter Data"

_Earth System Science Data, 2024_

## Author Comment (AC1)

**Responses to Reviewer's Comments**

2024-09-30

Dear Editor and Reviewers,

Revision of our manuscript ESSD-2024-277

Thank you for your comments and suggestions. We have made minor revisions to our paper (ESSD-2024-277 Global DEM Product Generation by Correcting ASTER GDEM Elevation with ICESat-2 Altimeter Data) according to the reviewer' suggestions (the changes have been highlighted). Referee comments are in **bold**, and the changed text is in *italics*.

Thank you again for your valuable comments and time.

Sincerely,

Prof. Huan Xie

College of Surveying and Geo-Informatics

Tongji University

1239 Siping Road, Shanghai, China,200092

Email: huanxie@tongji.edu.cn

Phone: ++86-21-65988851

**Reviewers Comments:**

**Q1. Can the authors numerically label the findings in the abstract, e.g., (1), (2), etc.?**
**Response:**
Thank you for your suggestion. We have revised the related descriptions.
In the abstract,
"*…The results from the validation comparison present that the elevation accuracy of IC2-GDEM is evidently superior to that of the ASTER GDEM product. The root-mean-square error (RMSE) reduction ratio of the corrected GDEM elevation is between 16% and 82%, and the average reduction ratio is about 47%. From the analysis of the different topographies and land covers, it was also found that this error reduction is effective even in areas with high topographic relief (>15 ° ) and high vegetation cover (>60%)…*"
has been revised to
"*"…The results from the validation comparison present that the elevation accuracy of IC2-GDEM is evidently superior to that of the ASTER GDEM product: 1) the root-mean-square error (RMSE) reduction ratio of the corrected GDEM elevation is between 16% and 82%, and the average reduction ratio is about 47%; 2) from the analysis of the different topographies and land covers, this error reduction is effective even in areas with high topographic relief (>15 ° ) and high vegetation cover (>60%)…*"

**Q2. L35-100: Some points within this paragraph need improvement, as follows:**
**First, the authors discuss the significance of refining ASTER GDEM but do not clearly explain what specific improvements or innovations this study introduces compared to previous studies.**
**Second, the paragraph starts by emphasizing the importance of high-quality DEMs, but the latter part seems to drift towards specific technical details about the ASTER GDEM correction method without clearly linking these details back to the broader impact or significance.**
**I found that the authors assumed that integrating ASTER GDEM with other DEM products will inherently lead to better outcomes but did not provide evidence or references to justify this assumption. Please revisit and address this point carefully.**
**I found phrases such as ＂high-accuracy global control point dataset＂ and ＂automatic processing scheme＂ are used without clear definitions or explanations of what makes them superior or innovative. This is very important for the product＇s validation in this work.**
**The significance of the study is stated multiple times (e.g., ＂of great significance,＂ ＂beneficial supplement＂), but without concrete examples or data to support these claims, the statements lack impact.**
**In addition, the literature review conducted on the use of remote-sensing DEMs Earth science research and scientific applications, including hydrological modeling, climate change research, natural hazard assessment, and ecosystem management was not well reviewed, suggesting accuracy of watershed delineation (10.1016/j.ejrh.2022.101282), flood risk assessment (10.3389/fenvs.2023.1304845), water resources management (10.1016/j.scitotenv.2024.174289 and 10.1007/s00382-024-07319-7), disaster preparedness (10.1109/jstars.2024.3380514), and promote human resilience for coastal communities (10.1016/j.jenvman.2024.121375).**

**Response:**

Thank you for your detail suggestions. Maybe the descriptions in the paper are not clear and sufficient enough. We have revised the relative descriptions and added some references in the Introduction, as follows:

*"In climate change research, high-quality DEMs are vital for accurate watershed delineation, disaster preparedness, promoting human resilience in coastal communities, modeling glacier dynamics, and assessing sea level rise impacts, etc., thereby improving predictive accuracy and informing mitigation strategies (Fan et al., 2022; Cook et al., 2012; Thanh-Nhan-Duc et al., 2023, 2024a, 2024b)."*

Reference:

*Tran, T.-N.-D., Tapas, M. R., Do, S. K., Etheridge, R., and Lakshmi, V.: Investigating the impacts of climate change on hydroclimatic extremes in the Tar-Pamlico River basin, North Carolina, J. Environ. Manage., 363, 121375, https://doi.org/10.1016/j.jenvman.2024.121375, 2024a.*

*Tran, T.-N.-D., Nguyen, B. Q., Vo, N. D., Le, M.-H., Nguyen, Q.-D., Lakshmi, V., and Bolten, J. D.: Quantification of global Digital Elevation Model (DEM) − A case study of the newly released NASADEM for a river basin in Central Vietnam, J. Hydrol.: Reg. Stud., 45, 101282, https://doi.org/10.1016/j.ejrh.2022.101282, 2023.*

*Tran, T. N. D., Do, S. K., Nguyen, B. Q., Tran, V. N., Grodzka-Łukaszewska, M., Sinicyn, G., and Lakshmi, V.: Investigating the Future Flood and Drought Shifts in the Transboundary Srepok River Basin Using CMIP6 Projections, IEEE J. Sel. Top. Appl. Earth Obs. Remote Sens., 17, 7516-7529, 10.1109/JSTARS.2024.3380514, 2024b.*

*"Furthermore, some studies (Pham et al., 2018; Yamazaki et al., 2017; Crippen et al., 2016; Franks et al., 2020; Okolie and Smit, 2022) have also reported that diversified choices and complementary use strategies for GDEMs are still favored by many scholars and users. For instance, the data sources for the void filling of the Copernicus GLO-30 DEM consist of other DEMs, including the ASTER GDEM, SRTM DEMs, and several national DEMs (Del Rosario González-Moradas et al., 2023). Thus, obtaining more accurate ASTER GDEM elevation products is a great significance for advancing the research of the GDEM diversified choices and complementary use strategies."*

*"Researchers can use the enhanced ASTER GDEM in conjunction with other DEM products to cross-validate qualities, fill data gaps, and conduct multi-scale analyses (Del Rosario González-Moradas et al., 2023). This complementary use of multiple DEMs can lead to more reliable and comprehensive scientific discoveries, thereby improving the overall quality and reliability of geoscience research (Del Rosario González-Moradas et al., 2023; Ao et al., 2024). However, the previous studies have hardly focused on integrating ASTER GDEM and satellite laser altimeter data to generate and release a new enhanced ASTER GDEM product at a global scale."*

*"In this study, based on these models, we further optimized and designed an automatic processing scheme for correcting ASTER GDEM elevation. This scheme does no required human intervention during the processing and is suitable for large-scale data processing."*

*"Meanwhile, a high-accuracy global elevation control point dataset (elevation RMSE: 0.5-3 m within different topographies) (Li, et al., 2022; Xie, et al., 2021) and multiple local DEMs (LDEMs) with a high resolution and accuracy were used to validate the accuracy of the original and refined ASTER GDEM, including comparison of the elevation accuracy in areas with different geolocations, topographic relief, and vegetation cover. The related validation results at global scale will provide a beneficial supplement for the accuracy qualification of the latest ASTER GDEM with different geolocations, altitudes, topography, vegetation cover, etc."*

**Q3. The study specifically excludes polar regions from the correction process due to challenges like high variability in ice sheets and flow rates. This exclusion limits the global applicability of the IC2-GDEM product and leaves a coverage gap, particularly for researchers focused on polar studies.**

**Response:**
Thank you for your suggestions. Compared with other areas, ICESat-2 laser altimeter observations are denser in polar areas, making this data particularly suitable for generating polar DEMs. Previous studies have refined ICESat/ICESat-2 altimeter data to produce and release polar DEM products. Moreover, ASTER GDEM does not fully cover polar areas, and correcting elevations in these regions is challenging due to the high variability in ice sheets and flow rates. For the researchers focused on polar studies, they can directly use the polar DEM production mainly sourced from ICESat-2 laser altimeter data. Maybe the descriptions in the paper are not clear and sufficient enough. We have revised some descriptions.

*"Given the particularity of polar areas, namely the high variability of ice sheets and ice flow rates, and the dense observations of satellite altimeter data, etc., only the parts of the product within the land areas between 83 ˚N and 60 ˚S (except for polar areas) were corrected."*
has been revised to
*"Compared with other areas, ICESat-2 laser altimeter observations are denser in polar areas, making this data particularly suitable for generating polar DEMs. Previous studies have refined ICESat/ICESat-2 altimeter data to produce and release polar DEM products. Moreover, ASTER GDEM does not fully cover polar areas, and correcting elevations in these regions is challenging due to the high variability in ice sheets and flow rates. Therefore, only the parts of the product within the land areas between 83 ˚N and 60 ˚S (except for polar areas) were corrected."*

**Q4. The potential for temporal inconsistencies between the ASTER GDEM data and the more recent ICESat-2 data is not fully discussed. In dynamic landscapes, such as areas experiencing rapid coastal erosion or land use changes, these temporal discrepancies could lead to inaccuracies in the corrected DEM, which the authors did not quantify or address adequately. Please revisit and provide reasonable discussions to address this point.**
**Response:**

Thank you for your suggestions. There is a potential for temporal inconsistencies between the ASTER GDEM data and the more recent ICESat-2 data, which may lead to inaccuracies in the corrected DEM in dynamic landscapes. To this end, we selected multiple areas located near the coast in the validation areas for evaluating the performance of DEM elevation correction, including the coast of the Netherlands, the coast of Australia, the islands of New Zealand, the west coast of the United States and the island of Hawaii, etc. From the results, it can be found that the corrected DEM elevations still have an obvious accuracy improvement. Maybe the descriptions in the paper are not clear and sufficient enough. We have added some descriptions.

In the section 4.4,

*"...Moreover, the temporal differences between the ICESat-2 data survey and ASTER GDEM collection may lead to elevation inconsistencies, especially in extremely dynamic landscapes (e.g., coastal erosion areas). To this end, for the validation areas, we selected multiple areas located near the coast for further evaluating the performance of DEM elevation correction in this kind of landscape, including the coast of the Netherlands, the coast of Australia, the islands of New Zealand, the west coast of the United States and the island of Hawaii, etc., as shown in Figure 2. From the validation results (as shown in Figure 8), it can be found that the corrected DEM elevations still have an obvious accuracy improvement. The main reason for this phenomenon is that the discrepancy caused by the change between the collection times of the two data sets is generally lower than the error in the ASTER GDEM elevation (about 10 m)"*

has been added.

**Q5. The authors acknowledged that the density of ICESat-2 observations varies significantly with latitude, but it does not thoroughly investigate how this variation impacts the accuracy of the DEM corrections. In low-latitude regions, where ICESat-2 data are sparser, the correction results might be less reliable, a factor that needs more detailed examination.**
**Response:**
Thank you for your suggestions. The data density of the ICESat-2 laser altimeter data varies greatly in different latitude regions, and the data density of the ICESat-2 laser altimeter data in the low-latitude regions is generally less than that in the high-latitude regions. From the corrected results, as shown in Fig. 7, we found that there is no significant difference in the DEM elevation correction errors between low-latitude and high-latitude areas. There are two main reasons causing this phenomenon. First, the DEM elevation correction model has a good generalization within different spatial distributions (Li et al., 2023). Second, in the generation of the GDEM elevation deviation, extracting the seed points (sourced from ICESat-2 laser altimeter data) of the model has been designed multiple strategies. These strategies ensure to obtain the even distribution of the seed points for further weakening the impact of the differences of density of the ICESat-2 laser altimeter data. Maybe the descriptions in the paper are not clear and sufficient enough. We have revised and added some descriptions.

After "*Similar to the analysis results in Figure 6, in the different latitude regions, it is also apparent that the systematic error and standard deviation after ASTER GDEM elevation correction are significantly reduced, compared with the original ASTER GDEM elevation, as displayed in Figure 7. Notably, the data density of the ICESat-2 laser altimeter data varies greatly in different latitude regions,*

*and the data density of the ICESat-2 laser altimeter data in the low-latitude regions is generally less than that in the high-latitude regions (Neuenschwander and Pitts, 2019; Markus et al., 2017). However, the errors of the ASTER GDEM elevation corrections in low-latitude regions (especially near the equator) are no larger than those in other latitude regions, which indicates that the correction results are less affected by the difference in the data density of the ICESat-2 laser altimeter data.*"

"*There are two main reasons for the observed phenomenon in DEM elevation correction. First, the DEM elevation correction model generalizes well across various spatial distributions (Li et al., 2023). Second, multiple strategies have been designed for extracting seed points from ICESat-2 laser altimeter data during the generation of GDEM elevation deviation. These strategies ensure an even distribution of seed points, thereby mitigating the impact of varying densities in the ICESat-2 laser altimeter data.*"
has been added.

**Q6. The authors briefly mention the challenges posed by dynamic landscapes, where changes between the times of data collection could lead to inconsistencies. However, it does not provide a detailed analysis or propose methods to mitigate these issues, which is crucial for applications in rapidly changing environments.**

**In general, please separate the Discussion from the Conclusion section and provide a more in-depth discussion based on qualitative results.**

**Q7. Please include a section on limitations and future work.**

**Response:**

Thank you for your suggestions. There are two strategies for weakening the impact of this kind of inconsistency. First, using repeated observations from the ICESat-2 satellite to analyze the areas with the elevation potential changes can qualify the application potential of IC2-GDEM within the dynamic landscapes. Second, integrating IC2-GDEM with other elevation data (closer to the ASTER GDEM data collection time) can reduce deviations in the analysis results within the dynamic landscapes. Maybe the descriptions in the paper are not clear and sufficient enough. We have separated a section on limitations and future work, and added some more in-depth discussions in this section.

"*4.4 Limitations and Future Work*
*From the above analysis, while the IC2-GDEM demonstrates significant improvements in elevation accuracy, its quality of elevation correction is notably limited in the areas with steep slopes or high vegetation cover index. For research or applications with high-quality elevation requirements, this limitation can be quantified and identified through topographic relief calculations or in combination with vegetation cover data. Moreover, the temporal differences between the ICESat-2 data survey and ASTER GDEM collection may lead to elevation inconsistencies, especially in extremely dynamic landscapes (e.g., coastal erosion areas). To this end, for the validation areas, we selected multiple areas located near the coast for further evaluating the performance of DEM elevation correction in this kind of landscape, including the coast of the Netherlands, the coast of Australia, the islands of New Zealand, the west coast of the United States and the island of Hawaii, etc., as shown in figure 2. Validation results (Figure 8) show that the corrected DEM elevations still exhibit significant accuracy improvements. This phenomenon is because the temporal discrepancy between the datasets is generally smaller than the ASTER GDEM elevation error (about 10 m). To further mitigate these inconsistencies, two main strategies are proposed. First, using repeated ICESat-2 observations to analyze areas with potential elevation changes can assess the application potential of IC2-GDEM in dynamic landscapes.*

*Second, integrating IC2-GDEM with other elevation data collected closer to the ASTER GDEM collection time can reduce analysis deviations in dynamic landscapes. In future work, we will identify areas with elevation changes using the above strategies and integrate other GDEMs to further enhance IC2-GDEM elevation quality in dynamic landscapes."* has been added.

**Q8. In the conclusion, please highlight the main findings with a brief description (suggest highlighting qualitative results), but please keep them short, direct, and concise. The current form is lengthy and difficult to follow.**

**Response:**

Thank you for your suggestions. We have revised the conclusion.

*"In this study, we introduce a new open-source dataset, named IC2-GDEM. IC2-GDEM is generated by directly refining the ASTER GDEM elevation with ICESat-2 altimeter data, as well as FROM-GLC10 data and GFCC30TC data. This strategy leverages existing datasets and enhances them with additional data and it is cost-effective for improving elevation accuracy, which is particularly beneficial for applications in developing regions where resources to obtain high-accuracy DEMs may be limited.*

*The elevation quality of IC2-GDEM has been evaluated at a global scale and in multiple local regions by using other laser/LiDAR data from satellite or airborne platforms. From the evaluation results for the different geolocations at a global scale, it was found that the correction results are less affected by the difference in the data density of the ICESat-2 laser altimeter data. ICESat-2 observations at higher-latitude areas are denser than those at lower-latitude areas. However, the errors of the IC2-GDEM elevation in low-latitude regions are no larger than the errors in high-latitude regions. From the evaluation results for the different topographies, the ASTER GDEM after elevation correction shows an obvious improvement of elevation quality for the different topographies, even in the mountainous areas with higher slopes. A similar conclusion was also found from the evaluation of the different land covers. These analysis results show that the IC2-GDEM presents a superior elevation quality at a global scale.*

*IC2-GDEM is expected to promote seamless integration with the historical datasets of ASTER GDEM, which is essential for longitudinal studies of long-term environmental changes, land use dynamics, and climate impacts. As a dataset for exploring the quality improvement of GDEM sourced from optical imaging, IC2-GDEM can serve as a new complementary data source for other GDEMs, such as SRTM, and Copernicus DEM. Researchers can combine IC2-GDEM with other DEMs to cross-validate qualities, fill data gaps, and conduct multi-scale analyses for earth science studies such as Flood risk, climate change, etc. with more reliable and comprehensive scientific discoveries."*

has been revised to

*"In this study, we introduce a new open-source dataset, named IC2-GDEM. IC2-GDEM is generated by directly refining the ASTER GDEM elevation with ICESat-2 altimeter data, as well as FROM-GLC10 data and GFCC30TC data. This strategy provides a cost-effective way to improve elevation accuracy, especially beneficial for developing regions with limited resources for high-accuracy DEMs. After validating at a global scale, IC2-GDEM presents a superior elevation quality and application potentials: 1) be less affected by the difference in the data density of the ICESat-2 laser altimeter data; 2) an obvious improvement of elevation quality for the different topographies, even in the mountainous areas with higher slopes; 3) a clear improvement in elevation quality was observed across various land covers, including areas with dense vegetation cover; 4) be expected to promote seamless integration with the historical datasets of ASTER GDEM for studying longitudinal studies of long-term environmental changes, land use dynamics, and climate impacts; 5) as a new complementary data*

*source for other GDEMs, such as SRTM, and Copernicus DEM to cross-validate qualities, fill data gaps and conduct multi-scale analyses.*"

---

## Author Comment (AC2)

**Reviewers Comments:**

**The paper is well written, and I have only a few minor suggestions:**

Thank you for your affirmation about our research and paper.

1) **In section 3.4 (GDEM Elevation Correction), the part on random forest regression could be expanded with more details, such as parameter selection, and supported by additional references. Additionally, I couldn't find the random forest in Figure 3, which raises some curiosity about its role in the overall correction process.**

**Response:**

Thank you for your suggestion. We have revised some descriptions to present more details about random forest regression. Meanwhile, we have revised the Fig. 3.

[Figure]

In the Section 3.4,

"*The random forest regression function (fitrensemble) of the MATLAB platform was directly used for its processing efficiency and compatibility, and its recommended/default setting values (including the number of trees was 100, etc.) were adopted.*"

has been revised to

"*The random forest regression function (fitrensemble) of the MATLAB platform was directly used for its processing efficiency and compatibility. The function can be viewed and downloaded via https://ww2.mathworks.cn/help/stats/fitrensemble.html. The method of function was selected to bootstrap aggregation (bagging, random forest) (Breiman, 2001). For this method, we adopted the recommended/default setting values for the parameter selection, including the tree number (100), learners (tree), etc.*"

*Breiman, L.: Random Forests, Machine Learning, 45, 5-32, 10.1023/A:1010933404324, 2001.*

**2)** **In section 4.1 or the abstract, it would be helpful to include more specific details about the new data, such as the spatial and temporal resolution. You might also consider creating a table summarizing the characteristics of the input and output data to allow users to quickly reference these features.**

**Response:**
Thank you for your suggestion. Maybe our description was not detailed and clear enough. We have added more specific details about the new data.

In the Section 4.1,
After "*Based on the presented scheme, we corrected the ASTER GDEM elevation with ICESat-2 altimeter data and then stored the corrected ASTER GDEM product (IC2-GDEM) in the GeoTIFF format (.tif) with the same projection and datum as the ASTER GDEM (Xie et al., 2024).*",
"*The resolution of IC2-GDEM is about 30 m (grid). The survey date of its main source data is between 2000-2013, and its elevation correction source is from ICESat-2 laser altimeter data with the survey date from 2018-2022. More details about the data source of the IC2-GDEM are listed in Table 2.*"
has been added

*Table 2 Characteristics of input and output data*

| Data Type | Data name | Resolution | Description |
|---|---|---|---|
| DEM data | ASTER GDEM V3 | ~30 m (grid) | Input data, main source data, survey date: 2000 - 2013 |
| Satellite laser altimeter data | ICESat-2 ATL08 | ~100 m (along the ground track) | Input data, main source data, survey date: 2018 - 2022 |
| Landcover data | FROM-GLC10 | ~10 m (grid) | Input data, auxiliary data, survey date: 2017 |
| Vegetation cover index data | GFCC30TC | ~30 m (grid) | Input data, auxiliary data, survey date: 2015 |
| DEM data | IC2-GDEM | ~30 m (grid) | Output data |

3) **The discussion in section 4 is somewhat limited, especially in sections 4.1 and 4.2, where it primarily presents results. It might be beneficial to discuss why the Corrected ASTER DEM Product showed greater improvement in Europe compared to other regions. Also, it seems there is no LDEM data for Asia and Africa, which could introduce uncertainties in validation—this could be worth discussing as well.**

**Response:**
Thank you for your suggestion. It's a beneficial discussion for our study and paper. We have added the related discussion.

In the Section 4.2,
After "*After the elevation correction, the errors of the ASTER GDEM elevation are reduced by more than 45%, and the maximum reduction ratio exceeds 70%.*"
has been revised to
"*There are two main reasons for this the inconsistency in elevation accuracy improvement among the different continents. The first is that there are significant differences in the quality of ASTER GDEMs across continents. The second is that the topographic relief and the landcovers across continents are different. These influencing factors are also the important evaluation attributes in the correction model of DEM elevation.*"

In the Section 4.3,
After the 5th paragraph,
*"The absence of LDEM data in Asia and Africa (difficult to obtain for free) is a noteworthy consideration. The lack of such data in these continents could introduce uncertainties, as the validation relies on the available LDEM data. To reduce the impact of these uncertainties, we also used the other satellite laser altimeter data with global coverage, i.e., ICESat altimeter data, as another source of the validation data in this study. The resolution and accuracy of the ICESat altimeter data may not match those of LDEM data. This means that the validation of finer topographic details in Asia and Africa may not fully account. Despite this gap, the similar validation accuracy between the two data is advantageous, particularly given the average reduction ratios close to 47%. This similarity suggests that ICESat altimeter data can serve as a reliable alternative and a beneficial supplement for areas lacking LDEM data."*
has been added.

---

## Author Comment (AC3)

**Reviewers Comments:**

The manuscript describes the correction of ASTER GDEM data with ICESat-2 data to create a product with a lower RMSE (according to a validation dataset). The ASTER GDEM is a widely used dataset and improving the accuracy is a useful exercise. The journal is the right venue to publish an open-source dataset.

The manuscript is mostly written in correct English and grammar – but please refrain from using etc. The abstract alone contains three occasions using etc. and several others throughout the manuscript. Either it is important enough to spell out – then list the additional points. If it is not important, there is no etc. needed. The term adds unnecessary ambiguity.

I can mostly follow the manuscript and reasoning, but have some comments. I understand that this article has seen previous reviews. I suggest that some of these are added as caveats or critical thoughts.

**Response:**

Thanks for your affirmation about our research and paper. We have checked the term "etc." in the whole paper and revised the related descriptions. According to your suggestions, we have revised the manuscript item by item and given a response to each comment.

1) Dataset description. A more detailed description of the ASTER GDEM is necessary. What is the time frame of acquisition? Is it reasonable to use an ICESat2 dataset to correct the data (ICESat2 likely postdates some of the scenes used in the generation of ASTER GDEM). Same with the validation dataset: The ICESat data likely predates the ASTER GDEM scenes. While it is not likely that the large number of validation points have changed and I don' t think there is an impact on the statistics – but it will be useful to give these relevant information and a word of caution. The years of the lidar DEMs are listed.

**Response:**

Thank you for your suggestion. We have added the information of the relevant data.

**In the Section 4.1,**

| Data Type                         | Data name      | Resolution                         | Description                                           |
|-----------------------------------|----------------|------------------------------------|--------------------------------------------------------------|
| DEM data                          | ASTER GDEM V3  | ~30 m (grid)                       | Input data, main source
data, survey date: 2000 -
2013 |
| Satellite laser
altimeter data | ICESat-2 ATL08 | ~100 m (along the
ground track) | Input data, main source
data, survey date: 2018 -
2022 |
| Landcover data                    | FROM-GLC10     | ~10 m (grid)                       | Input data, auxiliary
data,
survey date: 2017          |
| Vegetation cover index data       | GFCC30TC       | ~30 m (grid)                       | Input data, auxiliary
data, survey date: 2015             |
| DEM data                          | IC2-GDEM       | ~30 m (grid)                       | Output data                                                  |

**Table 2 Characteristics of input and output data**

has been added.

In the Section 4.2,

"By using the HAGECPD data, we validated and then contrasted the accuracy of the original and corrected ASTER GDEM elevation..."

has been revised to

"By using the HAGECPD data (survey date: 2003-2009), we validated and then contrasted the accuracy of the original and corrected ASTER GDEM elevation..."

2) While the training and validation dataset are somewhat independent (training: ICESat2, validation: ICESat about 10-15 years earlier), they both exhibit the same data characteristic. I have no concerns about the data quality, but it is not an unbiased validation. The geographic location points are not the same, but both data are point measurements (albeit taken with different instruments).

**Response:**

Thank you for your affirmation about our data quality. Our validation is not an absolute unbiased validation and it is difficult to achieve an absolute unbiased validation using high-accuracy data at a global scale. To reduce this impact, we adopted a comprehensive strategy, i.e., in addition to ICESat laser altimeter data, we also used LDEM to analyze ASTER GDEM before and after elevation correction. To describe clearer, we have revised the relevant descriptions.

**In the 1st paragraph of Section 2.4,**

After "Two different kinds of validation data were used in this study, according to the type of survey platform (i.e., satellite and airborne platforms)."

"We adopted this strategy for validating ASTER GDEM before and after elevation correction more comprehensively, in order to reduce the impact of biased validation." has been added.

3) I may have missed it, but how are the border effects of the individually-adjusted tiles treated? Each 1-degree tile is calibrated (or trained) individually and the adjustment parameters may be different than the neighboring parameters. This may cause (or not) a small offset at the boundaries of the tile. Initially, I thought there is a feathering approach used with a buffer (equations 1 to 5), but I am not certain that this point is clearly illustrated or explained. This is section 3.4.

**Response:**

Thank you for your suggestions. Yes, we used a buffer strategy to reduce the impact of the boundary, as shown in equations 1 to 5. The strategy we adopted is to expand the boundaries of the central DEM (the processing DEM) by using data from neighboring DEMs. The strategy described in Section 3.4 is to expand the area for selecting training samples in order to ensure that there are sufficient samples when the number of training samples is too low. Perhaps our description was not clear enough. To describe clearer, we have revised the relevant descriptions.

**The 1st paragraph before equations 1,**

"To address these challenges, it was necessary to expand the area around the DEM file to be corrected. As shown in part A of Figure 3, the expanded area includes two types..." has been revised to

"To address these challenges, it was necessary to expand the area around the DEM file to be corrected. We adopted a strategy that is to expand the boundaries of the central DEM (the processing DEM) by using data from neighboring DEMs. As shown in part A of Figure 3, the expanded area includes two types..."

**In the Section 3.4,**

"Moreover, the training model had a minimum requirement for the seed points' number within the modeling area."

**has been revised to**

"Moreover, when the number of training samples is too low, it needs to expand the area (i.e., the modeling area) for selecting training samples in order to ensure that there are sufficient training samples."

4) Where is the list of attributes that are trained with RF? Is there an attribute importance list that describe the usefulness of these parameters. I see the description of the "GDEM Elevation Evaluation Attribute Set" in 3.2 and that is useful. It is not clear what is contained in the elevation correction model (e.g., is this using Nuth and Kaeaeb to make horizontal adjustment or is this just a z component?)

**Response:**

Thank you for your suggestion.

In our previous study, we gave the details of the evaluation attributes in the regression model. The evaluation attribute set in the model includes topography, surface coverage, spatial distribution, and datasource quality. Meanwhile, we also evaluated the performance of each attribute. Our model just focuses on correcting DEM elevation, i.e., z component. Maybe we described the model as not clear. We have added some descriptions.

In the last paragraph of Section 3.2,

"The evaluation attribute set in the model includes topography, surface coverage, spatial distribution, and data-source quality (Li et al., 2023a)." has been added.

**In the last paragraph of Section 3.4,**

"After obtaining the model, all the DEM elevations were corrected by the model, and then a new DEM file was generated with the same format."

**has been revised to**

"After obtaining the model, all the DEM elevations (i.e., z component) were corrected by the model, and then a new DEM file was generated with the same format."

In our previous study:

Li, B., Xie, H., Tong, X., Tang, H., and Liu, S.: A Global-Scale DEM Elevation Correction Model Using ICESat-2 Laser Altimetry Data, IEEE Trans. Geosci. Remote Sens., 61, 1-15. Table Evaluation attributes in the regression model

| NO . | Factor                                      | Symbol                                              | Description                                                             |
|-------------|---------------------------------------------|-----------------------------------------------------|-------------------------------------------------------------------------|
| 1           |                                             | α                                                   | Aspect                                                                  |
| 2           |                                             | heta                                                | Slope                                                                   |
| 3           |                                             | $\theta_c$                                          | CV of slope                                                             |
| 4           | Toroostration                               | $	heta_	heta$                                       | Slope of slope                                                          |
| 5           | (Sland and ann act for turner)              | $lpha_	heta$                                        | Aspect of slope                                                         |
| 6           | (slope and aspect jeatures)                 | $	heta_{max}$                                       | Maximum slope                                                           |
| 7           | -                                           | $	heta_{STD}$                                       | STD of slope                                                            |
| 8           |                                             | $\Delta 	heta_{mean}$                               | $\theta - 	heta_{mean}$                                                 |
| 9           |                                             | $\Delta \theta_d$                                   | $\theta_{mean} - \theta_{min}$                                          |
| 10          | Topography
(Roughness and curvature)     | ξ                                                   | Roughness is calculated by
the DEM heights within a
3×3 window    |
| 11          |                                             | $\xi^{5	imes 5}$                                    | Roughness is calculated by
the DEM heights within a
5×5 window    |
| 12          |                                             | $C_{profc}$                                         | Profile curvature                                                       |
| 13          |                                             | $C_{planc}$                                         | Planar curvature                                                        |
| 14-15       | Topography
(Other statistical elevation) | $h_{mean}$ ( $h_{mean}^{5 \times 5}$ )              | Mean of the DEM heights within a $3 \times 3$ (5 $\times$ 5) window     |
| 16-17       |                                             | $h_{medain} \ (h_{medain}^{5 \times 5})$            | Median of the DEM heights within a $3 \times 3$ ( $5 \times 5$ ) window |
| 18-19       |                                             | $h_{min}$ $(h_{min}^{5 \times 5})$                  | Min of the DEM heights within a $3 \times 3$ ( $5 \times 5$ ) window    |
| 20-21       |                                             | $h_{max}$ $(h_{max}^{5 \times 5})$                  | Max of the DEM heights within a $3 \times 3$ (5 $\times 5$ ) window     |
| 22-23       |                                             | $\Delta h_{mean}~(\Delta h_{mean}^{5	imes 5})$      | $h - h_{mean}$ within a 3×3
(5×5) window                             |
| 24-25       |                                             | $\Delta h_{medain}~(\Delta h_{median}^{5 	imes 5})$ | $h - h_{median}$ within a 3×3
(5×5) window                           |
| 26-27       |                                             | $\Delta h_{min}~(\Delta h_{min}^{5 \times 5})$      | $h - h_{min}$ within a 3×3
(5×5) window                              |
| 28-29       |                                             | $\Delta h_{max} (\Delta h_{max}^{5 \times 5})$      | $h - h_{max}$ within a 3×3
(5×5) window                              |
| 30-31       |                                             | $\Delta h_{mm}~(\Delta h_{mm}^{5	imes 5})$          | $h_{max} - h_{min}$ within a 3×3
(5×5) window                        |
| 32-33       |                                             | $\Delta h_d ~(\Delta h_d^{5 	imes 5})$              | $h_{mean} - h_{min}$ within a 3×3
(5×5) window                       |
| 34          | Earth surface                               | $L_{cover}$                                         | Land cover                                                              |
| 35          |                                             | T index                                  | Vegetation cover index                                                  |

| 36 | Spatial Distribution and
Data-source Quality | $\Delta x$ | X-direction geographic location (relative value)    |
|----|-------------------------------------------------|------------|-----------------------------------------------------|
| 37 |                                                 | $\Delta y$ | Y-direction geographic
location (relative value) |
| 38 |                                                 | $\Delta z$ | Elevation (relative value)                          |
| 39 |                                                 | QA         | Data-source quality                                 |

Performance analysis of features evaluated in correction GDEM.

5) Along the same lines: Individually training each tile is useful and will allow to correct for local problems. The current description of the random forest training is a black-box approach – I did not see the parameters (or attributes) that are used for height adjustment (or I have missed them). Is this just the dH? In any case, it will be useful to include the adjustment parameter or vertical offset in a separate dataset. This will allow the user to see how much each tile has been adjusted. Most other global DEM datasets have additional quality data (e.g., Copernicus has the number of measurement or TanDEM-X pairs). The averaged adjusted dH value for each tile is a useful assessment metric.

**Response:**

Thank you for your suggestion. It's a beneficial suggestion. We have calculated the averaged adjusted dH value for each tile. We also upload the calculation information to the National Tibetan Plateau Data Center. Please see the Q7, figure 5 shows the distribution of the averaged adjusted dH value for each tile at a global scale.

6) I am wondering about the improved product. I can imagine that a post-processed ASTER GDEM has an reduced RMSE. But the core problem with optical data is the inherent noise (see also Purinton and Bookhagen, 2021). Even in vegetation-free areas, the correlation problems with optical data lead to lower quality DEMs. Is the noise level reduced by the random-forest filtering? A zoom in area of a characteristic area with sufficient detail would be useful (before and after correction). I point out that several studies now have shown that the Copernicus DEM is currently the best available DEM. It is difficult to compare optical and radar-based DEMs, because they measure different things. Different story again with a Lidar DEM.

**Response:**

Thank you for your suggestion.

Maybe we described the model as not clear. We analyzed the comparison of the original and corrected ASTER GDEM elevation error in vegetation-free areas, as shown in Figure 10 of the original paper. About the Copernicus DEM, your view is correct. Compared with the other global DEM data, the Copernicus DEM has the best quality in general. We have revised and added the relevant descriptions.

**In the Introduction,**

"Owing to the noise and anomalies resulting from the limitations inherent in optical imaging, the elevation quality of the ASTER GDEM is typically deemed to be lower than that of the other GDEMs for which the source is radar data (Meadows et al., 2024; Del Rosario González-Moradas et al., 2023)" has been revised to

"Owing to the noise and anomalies resulting from the limitations inherent in optical imaging, the elevation quality of the ASTER GDEM is typically deemed to be lower than that of the other GDEMs for which the source is radar data (Meadows et al., 2024; Del Rosario González-Moradas et al., 2023; Purinton and Bookhagen, 2021)"

"For instance, the data sources for the void filling of the Copernicus GLO-30 DEM consist of other DEMs..."

**has been revised to**

"For instance, the data sources for the void filling of the Copernicus GLO-30 DEM (currently the best available global DEM) consist of other DEMs..."

**In reference,**

"Purinton, B. and Bookhagen, B.: Beyond Vertical Point Accuracy: Assessing Inter-pixel Consistency in 30 m Global DEMs for the Arid Central Andes, Front. Earth Sci., 9, 2021." has been added.

In the 4th paragraph of the Section 4.3,

After "For the different land covers, the corrected GDEM elevation in the bare land areas shows the best elevation correction.", "This means that the DEM elevation correction model can reduce the impact caused by the inherent noise of optical data." has been added

In our previous study, we focused on the performance of the elevation correction model and presented some examples of the detail about quality improvement of ASTER GDEM elevation before and after, as the following figure.